# Study on the Dynamic Evolution and Regional Differences of the Level of High-Quality Economic and Social Development in China

**Qiangsheng Mai** [1] , **Mengting Bai** [2],*** and Le Li** [2]

1    School of Accounting, Southwest Forestry University, Kunming 650224, China
2    School of Economics and Management, Southwest Forestry University, Kunming 650224, China
*    Correspondence: mengting6257@sohu.com

**Abstract:** Analyzing the dynamic evolution and regional differences of the level of high-quality economic and social development based on the perspective of long time can be beneficial for informing the effect of the economy in China. This study builds a development evaluation index system oriented to high-quality economic and social development in five directions. To deepen the analysis, TOPSIS entropy is used to measure the level of economic and social development of 31 provinces in China from 2000 to 2019, considering the temporal characteristics and evolution laws of four regions. Dagum's Gini coefficient method is also applied to measure regional differences and discover driving factors. There are three key results. First, the comprehensive development index runs well; however, the index of economic momentum, innovationability, and safety assurance differ significantly. There is obvious path dependence, demonstrating a spatial gradient, with development that is high in the east, moderate in the central part, and low in the west. Second, the trend of the comprehensive development index of each province is gratifying, but there are obvious differences in the three dimensions of economic momentum, innovation ability, and security index. Third, the overall development coordination of the four regions is weak—that is, the levels of economic and social development oriented to high-quality intensified differentiation; it will be difficult to close the regional gap in the short term, given the development heterogeneity among regions. To promote the formation of a new, balanced pattern of regional development of high quality with complementary advantages, the four regions should be targeted in a double cycle of implementing regional development strategies and enhancing development coordination. Attention should also be paid to the growth pole function of advanced regions, complementing the advantages of other regions, and ultimately promoting high-quality development jointly.

**Keywords:** high quality development; regional differences; TOPSIS entropy; Dagum's Gini coefficient





## 1. Introduction

The improvement of economy development quality is not only the concern of the development of the country, but also an important source of stability in human society [1]. Economy-dependent programs have been embarked by successive governments as means to maintain long-term development and stability in China. High-quality development is the goal of the 14th Five-Year Plan, as well as a key economic and social development theme for China in the longer term, related to the overall situation of the country's socialist modernization. Steadfastly following the path of high-quality development is a general requirement, not only for many aspects of economic and social development, but also for regional development [2]. This requirement not only concerns a certain time and concept, but the direction must be adhered to in the long term [3]. Since the 19th Party Congress report proposed that China's economy has shifted from the stage of high-speed growth to that of high-quality development, academics have engaged in in-depth discussions and

analyses on the connotations of high-quality economic development from different per-spectives, particularly in the dimensions of scientific and technological innovations, market mechanisms, government regulations, and the business environment [4]. The connotation of such development remains unclear, however, and the driving factors of regional differ-ences are explored even less frequently. High-quality economic and social development are the inevitable paths for solving the inadequate and uneven development and must have the corresponding theoretical guidance to reveal their basic laws [5]. The construction of an evaluation index system for high-quality economic and social development and conducting dynamic evolution analysis at both a comprehensive level and for regional differences can not only provide a theoretical basis for development, but also provide a practical reference for formulating regional development policies [6].

Taking into account the current stage characteristics of China's economic and social development, as well as the new requirements put forward in the 14th Five-Year Plan, this study constructs an evaluation index system for economic and social development, measures the development level of 31 provinces in China from 2000 to 2019. The analysis uses the entropy-weighted TOPSIS method, and analyzes the time-series characteristics and evolution patterns of the development of the four major regions, as well as the regional differences and driving factors using Dagum's Gini coefficient method. The goal is to make theoretical and practical contributions to the realization of high-quality economic and social development in the academic world.

## 2. Literature Review and Comments

Along with the new economic normal, economic and social developments are un-dergoing profound changes in the world [7]. High-quality development has become a fundamental requirement for governments all around the world to determine its ideas related to development, formulate economic policies, and implement macro-control in the present and future. Evidently, academics focus on the economic factors explaining the type of apparently "miracle" provisions. For this purpose, many scholars will use the framework of development economics, where each of a set of economic agents who are tied by demand or cost complementarities, decide either to stick to a "traditional technology" (small scale, constant returns, local sales only) [8] or to switch to a "modern technology"(large scale, increasing returns, exports) [9].

In recent years, most studies have focused on qualitative analysis of the connotations, influencing factors, and path choices of high-quality economic development, and the consistent conclusion is that their promotion is a natural choice to adapt to the changes in the tensions within China during the new era [10].

Attempts have been made to evaluate the quality of economic development and the imbalance of regional development quantitatively. Earlier studies have measured the level of regional economic development by a single representative indicator such as per capita GDP or GNP. For example, Gui and others [11] used per capita GDP as an indicator with spatial measures to study the spatial and temporal evolution of economic differences and influencing factors in the Yangtze River Economic Belt. Chao and others [12] used the annual data on China's GDP to present the trends of economic growth. They then used a non-linear model to explore the inherent linkage mechanism between environmental pollution and economic growth. Xiang and others [13] used the Dagum's Gini coefficient and its decomposition method to decompose the regional differences of green development efficiency of the chemical industry in the Economic Belt, and the coefficient of variation method and panel data regression model to test the convergence characteristics. Although these studies have achieved positive results, using a single indicator as the measurement method is obviously one-sided and has limitations, particularly as it cannot reveal the whole picture of high-quality economic development.

Given the depth of research, most scholars have tended to use a multiple evaluation index system. Feng and Guo [14] analyzed the connotations of high-quality economic development and constructed an evaluation system consisting of the five components

of economic vitality, innovation efficiency, green development, people's life, and social harmony. Li and others [15] constructed and empirically analyzed the index system for high-quality economic development of cities in the Yangtze River Economic Belt based on the five development concepts of innovation, coordination, green, openness, and sharing.

Regarding the research on high economic quality, academics have paid more attention to the differences in the quality of regional economic development, as well as to the continuous improvement of the evaluation index system [16]. Common methods include Dagum's Gini coefficient [17], coefficient of variation [18], and Theil's index [19]. Zhang and others [20] use the inter-provincial agricultural industry panel data from 2000 to 2019 and selected the three-stage super-efficiency slack-based measure data envelope analysis model to measure ACEE. Additionally, they then used the Dagum's Gini coefficient and the kernel density model to analyze the sources of regional differences in ACEE and the internal dynamic evolution. Yu and others [21] used the Theil index and a principal component analysis to measure the equity of government health expenditure and the health level of residents in the Chengdu–Chongqing economic circle. Such research methods focus on measuring the general evolutionary trend of the degree of regional economic differentiation, while some scholars have begun to focus on the application of spatial analysis methods such as GIS and ESDA [22,23]. Typical studies, such as those performed by He and others [24], have used a combination of GIS and exploratory spatial data analysis to examine the spatial and temporal evolution patterns of interprovincial high-quality development. Ye and Carroll [25] explored the spatial and temporal evolutionary characteristics of economic disparities in the context of high-quality development through a spatial econometric model.

Looking at the existing research literature, the few studies on high-quality development are biased toward the field of economics, while there is a serious lack of research on the combination of high-quality economic and social development. The current measurement dimensions and evaluation indicators vary greatly, and the measurement methods mostly use the entropy method, but it is difficult to avoid the unstable influence of extreme values. In addition, scholars have mostly focused on overall regional differences, but less on inter- and intra-regional differences, and it is difficult to reveal the evolutionary characteristics of high-quality economic and social development without a spatial perspective. This study constructs an evaluation index system for high-quality economic and social development based on the five major directions proposed in the Outline of the 14th Five-Year Plan of the National Economic and Social Development of the People's Republic of China and Vision 2035; the development level of 31 provinces in China from 2000 to 2019 is then measured using the entropy-weighted TOPSIS method; The time-series characteristics and evolution patterns of the development of the four major regions are then analyzed using radar maps and GIS maps. Dagum's Gini coefficient method is also used to analyze the regional differences and drivers of development to provide a reference for deepening research on high-quality economic and social development.

## 3. Research Program

### 3.1. Construction of the Evaluation Index System

The Outline of the 14th Five-Year Plan of the National Economic and Social Development of the People's Republic of China and the Vision 2035 (the 14th Five-Year Plan) sets forth five major directions for economic and social development. The first is economic development, including adherence to the new development concept, ensuring quality and efficiency on the basis of significant improvements to achieve sustainable and healthy economic development, keeping GDP growth rate in a reasonable range, and ensuring urban and rural regional development coordination is significantly enhanced. The second direction includes innovation-driven aspects, with adherence to innovation-driven development and comprehensive shaping of new advantages in development. This involves steadily increasing the proportion of research and development expenditure to GDP and achieving a significant increase in the level of advanced industrial base and modernization of the industrial chain. The third direction is the well-being of people's livelihood to achieve fuller

and higher quality employment, as well as the basic synchronization of income growth and economic growth, the obvious improvement of the distribution structure, the obvious increase in the level of equalization of basic public services, the continuous increase in the level of education for all, a more complete multi-level social security system, and a more complete health system. The fourth direction covers green ecological aspects, optimization of the pattern of territorial space development and protection, the effective green transformation of production and lifestyle, the reasonable allocation of energy resources, significant improvement in the efficiency of resource utilization, continuous reduction in the total emissions of major pollutants, improvement of the ecological environment, solidifying the ecological security barrier, and the significant improvement of the urban and rural habitat. The fifth and final direction focuses on safety and security, improving the ability of food, energy, and other areas of safe development, enhancing public safety and security capabilities, and building a higher level of peace in China.

This study takes these development goals and follows the principles of comprehensiveness, relevance, and operability, while considering the validity and availability of data, and then makes the corresponding substitutions with reference to the practices of other scholars to circumvent the problems of missing data caused by differences in statistical or newly included indicators [26,27]. The final construction of the evaluation index system of high-quality economic and social development, which consists of five dimensions and eighteen secondary indicators, including economic dynamics, innovation capability, people's welfare, green ecology, and security, is shown in Table 1.

**Table 1.** System of indicators for evaluating high-quality economic and social development.

| First Level Indicators | Secondary Indicators | Calculation Method | Unit | Properties |
| --- | --- | --- | --- | --- |
| economic dynamics | GDP growth ratio | based on constant price GDP | % | + |
| | total Labor Productivity | GDP/average number of employees | % | + |
| | resident Population urbanization ratio | direct access to numbers | % | + |
| innovation capability | growth rate of social investment in R&D | based on R&D funding | % | + |
| | accumulated number of patents granted | direct access to numbers | piece | + |
| | growth rate of per capital disposable income of urban residents | based on disposable income per inhabitant | % | + |
| | registered urban unemployment rate | direct access to numbers | % | - |
| people's welfare | average number of years of schooling for the population over 6 years old | total years of schooling/population | year | + |
| | consumer Price Index (CPI) | direct access to numbers | % | - |
| | number of practicing physicians per 1000 population | number of practitioners/Population | people/thousands | + |
| | number of urban workers' basic pension insurance participants | direct access to numbers | ten thousand person | + |
| | sulphur dioxide emissions | direct access to numbers | tons | - |
| | chemical oxygen demand (COD) | direct access to numbers | million tons | - |
| green ecology | general industrial solid waste generation | direct access to numbers | million tons | - |
| | general industrial solid waste integrated use volume | direct access to numbers | million tons | + |
| | forest cover | direct access to numbers | % | + |
| security | annual grain production | direct access to numbers | million tons | + |
| | annual production of electricity | direct access to numbers | billion kWh | + |

## 3.2. Research Methods

### 3.2.1. Entropy TOPSIS Method

The entropy TOPSIS technique is used in this study to gauge the degree of high-quality development of China's society and economy. The entropy-weighted TOPSIS approach is a multi-attribute decision-making technique with good dynamic and objective weighting characteristics that can raise the legitimacy of thorough multi-indicator evaluation. Based on how much the available data varies, it objectively assigns values to each indicator. Its

main idea is to rank the relative qualities of existing solutions in accordance with how closely the evaluation object resembles the ideal goal. The exact calculating steps are as follows.

1.　Direct computation will contain large errors due to the indicators' different magnitudes. In order to account for large fluctuations, the chosen raw data must be normalized.

$$Y_{ij} = \begin{cases} \frac{x_{ij} - min(x_{ij})}{max(x_{ij}) - min(x_{ij})}, & x_{ij} \text{ is a positive indicator} \\ \frac{max(x_{ij}) - x_{ij}}{max(x_{ij}) - min(x_{ij})}, & x_{ij} \text{ is a negative indicator} \end{cases} \tag{1}$$

$X_{ij}$ stands for the initial value of the *j*-th index of the *i*-th province, where *i* stands for a province and *j* for a measurement indication. $Y_{ij}$ is the outcome of dimensionless data; $Min(x_{ij})$ and $max(x_{ij})$ denote the index's minimal and maximum values, respectively.

2.　Calculate the share of the *i*-th province under the *j*-th indicator:

$$p_{ij} = \frac{x_{ij}}{\sum_{i=1}^{n} x_{ij}} \tag{2}$$

3.　Calculate the entropy of the *j*-th entropy value of the indicator:

$$e_j = -k \sum_{i=1}^{n} p_{ij} \ln(p_{ij}) \tag{3}$$

Here, $k = \frac{1}{\ln(n)}$, $k > 0$ and $e_j > 0$.

4.　Calculate the weights for each indicator:

$$W_j = \frac{1 - e_j}{\sum_{j=1}^{m} 1 - e_j} \tag{4}$$

where $w_j$ is the weight of the *j*-th indicator, *m* is the number of evaluation indicators, and $e_j$ denotes the entropy value of the *j*-th indicator.

5.　Define the weighting matrix:

$$T = (t_{ij})_{m \times n}, \ t_{ij} = w_j * y_{ij} \ (i = 1, 2, \dots, n; j = 1, 2, \dots, m) \tag{5}$$

6.　Determine the optimal $R_j^+$ and inferior solution's $R_j^-$:

$$R_j^+ = max(t_{1j}, t_{2j}, \dots, t_{nj}), \ R_j^- = min(t_{1j}, t_{2j}, \dots, t_{nj}) \tag{6}$$

7.　The Euclidean distance method was used to calculate the distance from the object of assessment to the optimal solution and the worst solution:

$$d_i^+ = \sqrt{\sum_{j=1}^{m} (R_j^+ - t_{ij})^2}, \ d_i^- = \sqrt{\sum_{j=1}^{m} (R_j^- - t_{ij})^2} \tag{7}$$

8.　Calculate the fit of each evaluation object to the positive and negative ideal solution, i.e., the composite index:

$$C_i = \frac{d_i^-}{d_i^- + d_i^+} \tag{8}$$

Among them, the relative proximity $C_i$ ranges between 0 and 1. The larger the value of $C_i$, the better the economic and social development level of the *i*-th province. Otherwise, the worse the economic and social development level of the *i*-th province.

### 3.2.2. Dagum's Gini Coefficient Method

The Dagum Gini coefficient approach is a tool for analyzing the causes of spatial inequalities and efficiently describing them. Dagum originally suggested the idea in 1997. To examine the differences in China's quality of economic and social growth, the authors use the Dagum Gini coefficient method, which is defined and calculated as follows:

$$G = \frac{\sum_{j=1}^{k} \sum_{h=1}^{k} \sum_{i=1}^{n_j} \sum_{r=1}^{n_h} |y_{ji} - y_{hr}|}{2n^2 y} \tag{9}$$

$$\overline{Y_h} \leq \cdots \overline{Y_j} \leq \cdots \overline{Y_k} \tag{10}$$

$K$ is the number of regional divisions, which is set to four in this case (that is, Eastern Region, Central Region, Western Region, and North Eastern Region). The 31 provinces are indicated by $n$. $Y_{ji}y_{hr}$ represents the degree of high quality economic and social development of a province $i$ ($r$) in a region $j$ ($h$). $\overline{Y}$ represents the overall average of all provinces' high-quality economic and social development. An increased Dagum's Gini coefficient denotes a more unequal state of social and economic development.

$$G_{ij} = \frac{\frac{1}{2\overline{Y}} \sum_{i=1}^{n_j} \sum_{r=1}^{n_j} |y_{ji} - y_{jr}|}{n_j^2} \tag{11}$$

$$G_w = \sum_{j=1}^{k} G_{jj} p_j s_j \tag{12}$$

$$G_{jh} = \frac{\sum_{i=1}^{n_j} \sum_{r=1}^{n_h} |y_{ji} - y_{hr}|}{n_j n_h (Y_j + Y_h)} \tag{13}$$

$$G_{nb} = \sum_{j=2}^{k} \sum_{h=1}^{j-1} G_{jh} (p_j s_h + p_h s_j) D_{jh} \tag{14}$$

$$G_t = \sum_{j=2}^{k} \sum_{h=1}^{j-1} G_{jh} (p_j s_h + p_h s_j) \left(1 - D_{jh}\right) \tag{15}$$

where the intra-regional contribution to variation is $G_w$, the inter-regional contribution to variation is $G_{nb}$, and the contribution of hyper-variable density is $G_t$, which can be used to decompose the Dagum Gini coefficient. $G = G_w + G_{nb} + G_t$ describes the relationship between the three elements. Equations (12), (14) and (15) are used, respectively, to express them. The Dagum Gini coefficient for the region $j$ is $G_{jj}$ in Equation (11). The inter-provincial Dagum Gini coefficient for provinces $j$ and $h$ is $G_{jh}$ in Equation (13).

$$D_{jh} = \frac{d_{jh} - p_{jh}}{d_{jh} + p_{jh}} \tag{16}$$

$$d_{jh} = \int_0^\infty dF_j(y) \int_0^y (y - x) dF_h(x) \tag{17}$$

$$p_{jh} = \int_0^\infty dF_h(y) \int_0^y (y - x) dF_j(x) \tag{18}$$

The $p_{jh}$ in Equation (18) is the hyper-variable first-order moment. It denotes the mathematical expectation of the sum of all $y_{hr} > y_{ji}$ sample values in provinces $j$ and $h$. $F_j$ ($F_h$) is the cumulative density distribution function for $j$ ($h$) provinces.

### 3.2.3. Data Sources

This study selects relevant data from 31 provinces in China (excluding Hong Kong, Macao, and Taiwan) from 2000 to 2019 to measure the level of quality economic and social development in China. The data are obtained from the China Statistical Yearbook, China Environmental Statistical Yearbook, China Population and Employment Statistical Yearbook, China Labor Statistical Yearbook, China Health and Health Statistical Yearbook,

and China Energy Statistical Yearbook, as well as the statistical yearbooks of various provinces (autonomous regions and municipalities directly under the Central Government) and the national economic and social development bulletins. Owing to missing data for a small number of indicators in Tibet, interpolation was used based on data from adjacent years.

## 4. Evolutionary Characteristics of Quality Level of Economic and Social Development

*4.1. Composite Index: Converging Trend. Sub-Indices: Diverging Trend*

According to Equations (1)–(8), the comprehensive index of China's historical high quality economic and social development level and each dimension index were measured; the results are shown in Figure 1. The values corresponding to the curves closer to the heart of the circle are smaller and increase as the curve spreads towards the periphery. As can be seen from Figure 1a, the part of the composite index curve near the center of the circle is sparse, and the off-center curve is denser, reflecting the convergent development trend for each province from 2000 to 2019; except for Jiangsu, Guangdong, and Zhejiang, which have a relatively high composite index, most of the provinces' economic and social development are neck and neck in terms of quality. However, there are still a few regions at a relatively low level, and the differences in development between regions do not show a significant reduction over the entire sample period, suggesting a persistent regional imbalance in the overall level of economic and social development. As can be seen from Figure 1b, the economic momentum of China's provinces is uneven, and there are two poles in the spatial distribution, namely, Beijing and Shanghai, and the economic momentum index varies widely by province. Hubei, Chongqing, Shaanxi, and Fujian, for example, have achieved relatively substantial growth in the past two decades and still have high potential for future development. As can be seen from Figure 1c, the unevenness of innovation capacity across provinces is more pronounced than for economic momentum, with Jiangsu, Zhejiang, and Guangdong having high innovation capacity indices, while the other provinces are at lower levels. As can be seen from Figure 1d, the livelihood well-being indices of all provinces show year-on-year improvement, with Beijing and Guangdong consistently displaying relatively high levels of development during the sample observation period, which is closely related to the good policy environment and economic strength. As can be seen from Figure 1e, the unevenness of the green ecological index is relatively low compared to the other dimensions, with all provinces showing rapid green ecological development, thanks to China's co-ordination of ecological protection and economic development to maximize ecological and economic benefits, as well as the rapid increase in the curbing of total emissions of major pollutants. As can be seen from Figure 1f, the security index is significantly higher in Heilongjiang, Shandong, and Henan provinces, mainly because these three provinces are large grain provinces and have made outstanding contributions to ensuring national food security.

The above analysis shows that the level of quality for economic and social development in China's provinces has been increasing year by year, but uneven development remains a persistent problem. Among the five dimensions, the imbalance in regional development is more pronounced in the innovation capacity and safety and security dimensions, followed by the economic dynamics dimension, while there is more balanced development in the livelihood and well-being and green ecology dimensions.

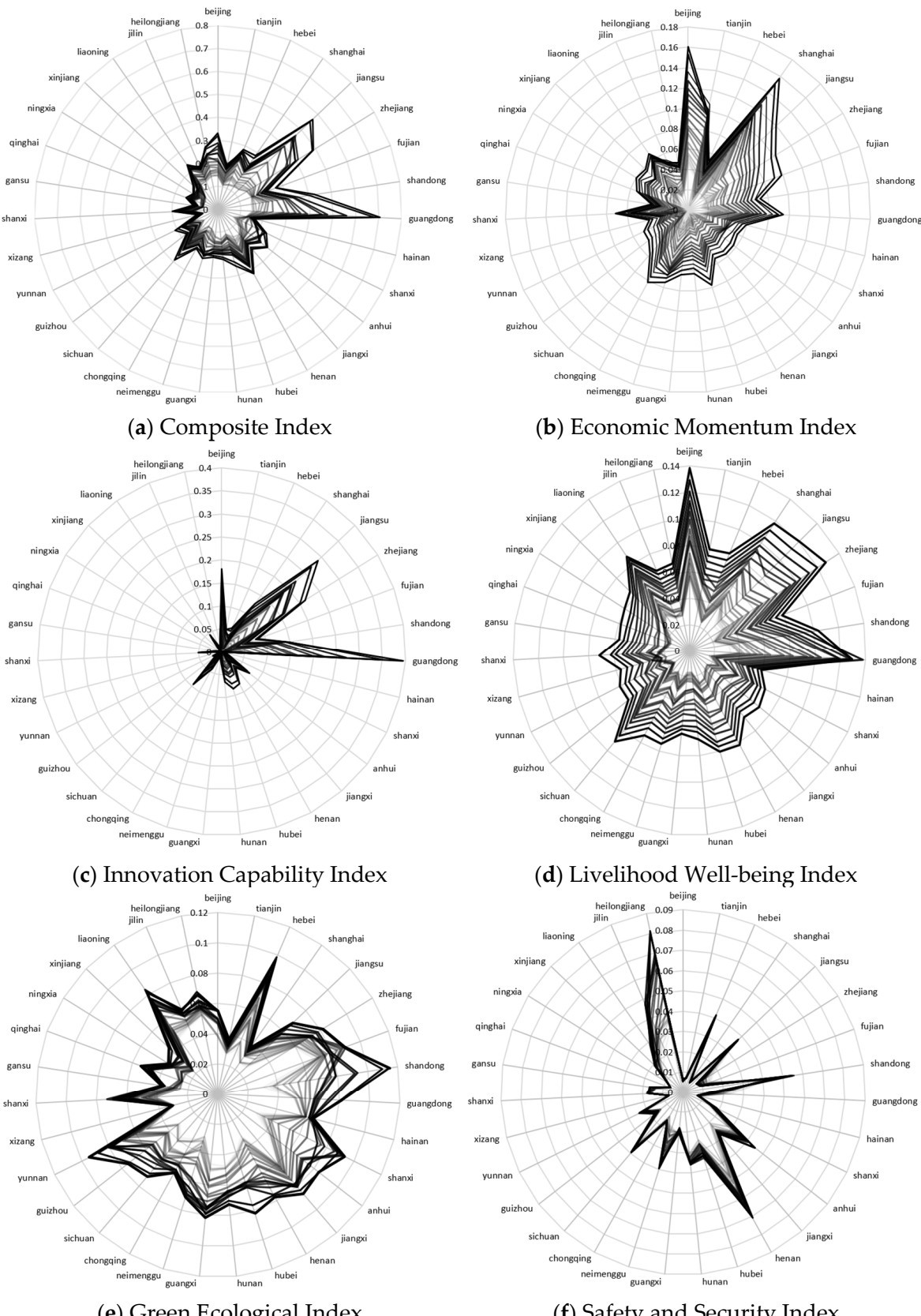

**Figure 1.** Comprehensive level of quality for economic and social development across five dimensional indices. Note: The lightest color corresponds to the indicator for each province in 2000, and the color of the corresponding curve deepens as the year increases.

*4.2. National Development Trend Continues, Inter-Provincial Development Path Dependence Remains*

This study selects four time cross-sections, 2000, 2006, 2012, and 2019, and classifies them into four categories-namely, (a) low level, (b) medium-low level, (c) medium-high level, and (d) high level-using the composite index of each year as the benchmark. ArcGIS 10.8 software was used to map the spatial distribution of high quality economic and social development levels. The results are shown in Figure 2.

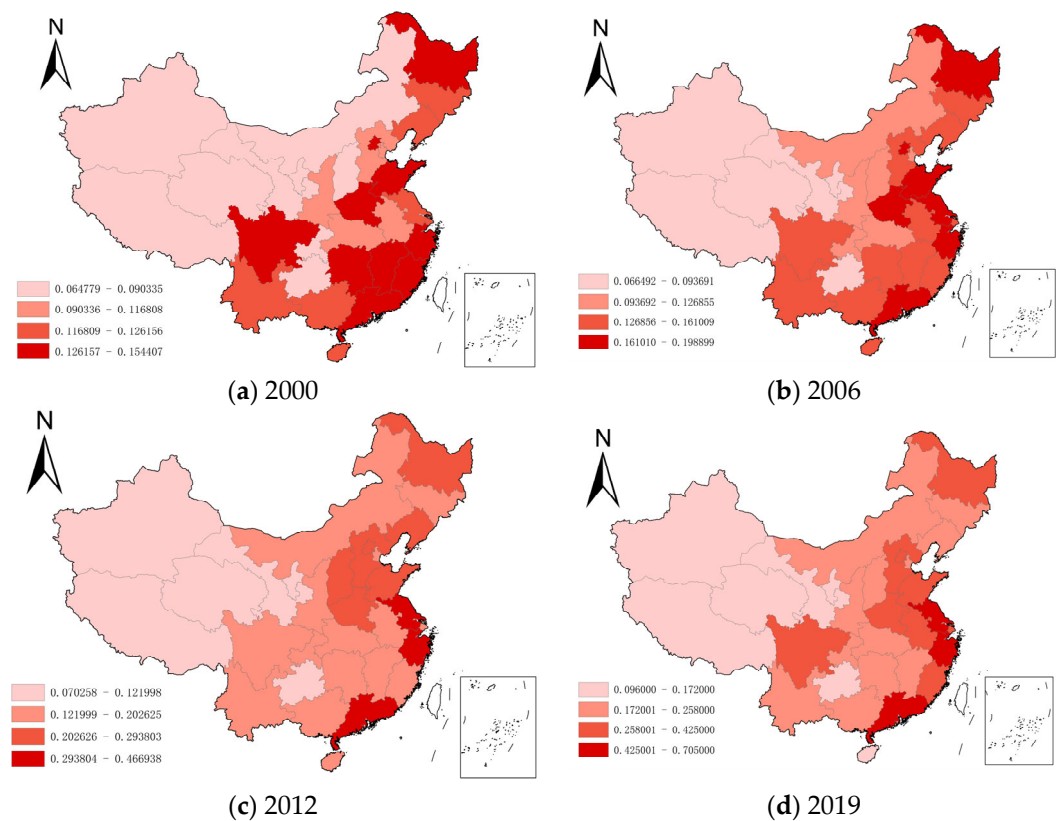

**Figure 2.** Spatial dynamic evolution of the quality of economic and social development.

As can be seen from Figure 2, there has been a general increase in the level of high-quality economic and social development over time, and the intervals in the legend for each year reflect this "leapfrogging" progress. In addition, the zones of high-quality development were initially scattered in the eastern and central regions, but have since been clustered in the eastern coastal regions, with Jiangsu and Guangdong becoming the leaders. The dual core dominance of Jiangsu and Guangdong has led to a significant polarization effect, with the development of the eastern coastal regions centered on the two provinces, and a widening gap separating the inland regions.

Looking at the spatial evolution characteristics, we can see that there are differences in the quality of economic and social development across the country, with most of the high level development areas concentrated in the eastern coastal areas. The overall pattern follows a gradient divergence of high in the east and low in the west, high in the south and low in the north. The evolutionary characteristics over time show that the inheritance of inter-provincial high quality economic and social development is significant; that is, the quality of development in each province is based on the evolution of the pattern over a longer period of time, showing obvious path dependence, so it is difficult to achieve a leap across levels in the short term. Over time, the spatial imbalance in China's development will continue, and the gap will continue to widen, making it more difficult to bridge the gap. In the dynamic evolution of China's development, it is worth noting that the composite index of the developed coastal regions has shown strong resilience and tension; it has not fallen

due to factors such as industrial restructuring, transformation, and upgrading. Overall, the country's development has followed a better development trend, but the problem of unbalanced regional development due to factors such as resource endowment, location conditions, and national policies still has not been effectively alleviated, but is instead gradually expanding. Both developed and less developed regions thus need to find their own weak links and focus on breakthroughs to achieve fuller quality development.

## 5. Difference in the Level of Quality Economic and Social Development

### 5.1. Overall Variance Fluctuating Upwards

To visually analyze the evolutionary trend of the difference in quality economic and social development in China, the trend of Dagum's Gini coefficient was measured and plotted according to Equations (9)–(18), as shown in Figure 3. The overall difference in the level of development in China from 2000 to 2019 shows a fluctuating increase, with Dagum's Gini coefficient rising from 0.141 in 2000 to 0.246 in 2019, with an average annual increase of 5.25%. This indicates a gradual widening of the difference over the sample period, but the Gini coefficient fluctuates between 0.14 and 0.25, within a reasonable threshold. (A Gini coefficient of income equal to 0.4 is a "warning line" according to the United Nations; if the Gini coefficient for income exceeds 0.4, it means that there is a gap in income distribution and inequity. This criterion also applies to the Gini coefficients for education, health, and housing, and each Gini coefficient is negatively correlated with its inequality—that is, the higher the Gini coefficient, the more unequal the corresponding distribution of resources, while, conversely, the lower the coefficient, the more equitable the distribution.) Dagum's Gini coefficient evolved in two phases, the first being from 2000 to 2012, when the Gini coefficient showed a flattening U-shaped trend, reaching a low of 0.135 in 2004. In the second phase, from 2012 to 2019, the Gini coefficient rose in an M shape, reaching its maximum level of variation over the sample period in 2018, with the overall variation rising through three peaks, the first two with an average annual growth rate of 7.67% and the final two with an average annual growth rate of 4.49%. It is easy to see that the increase of the Dagum's Gini coefficient has increased in speed, but does not show a strict increasing trend, and the regional development synergy remains weak. For the overall disparities to be sustained and effectively mitigated, there is an urgent need to highlight the leading role of national regional development strategies to effectively narrow regional gaps, alleviate regional economic development conflicts, and enhance inter-regional coordination.

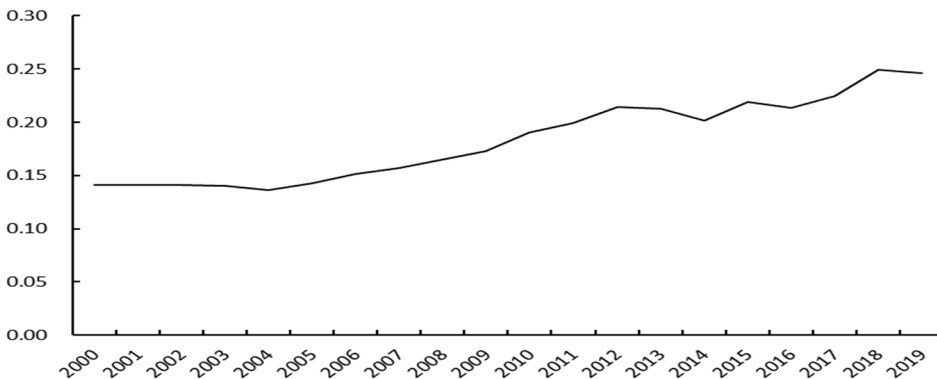

**Figure 3.** Overall differences in the level of quality economic and social development.

### 5.2. Growing Intra-Regional Differences, Particularly in the East

The trends in the evolution of intra-regional differences in terms of the level of quality economic and social development are shown in Figure 4. Overall, the four regions are at different levels of development quality and show significantly different evolutionary trends. From 2000 to 2007, the differences within the four major regions tended to stabilize and ranked as northeast < eastern < central < western. During the period 2007–2019, the four major regions began to show a clearly differentiated trend and ranked as northeast

< central < western < eastern. The Gini coefficient for the central region decreased over the sample period in a fluctuating manner, with a steady overall trend. The reason for this is that there is less economic cooperation between cities in this region; the inter-city economic links are not strong, making it difficult to encourage interconnected interaction between cities; The gap between the levels of economic and social development in the region is not large. Dagum's Gini coefficients for the Northeast and West remain flat, with annual growth rates of 0.46% and 0.49%, respectively. Dagum's Gini coefficient has risen sharply in the eastern region, with an annual growth rate of 2%, mainly because some provinces and municipalities in the eastern region have seen their level of economic and social development improve significantly faster than other provinces and municipalities in the region, widening inter-regional differences.

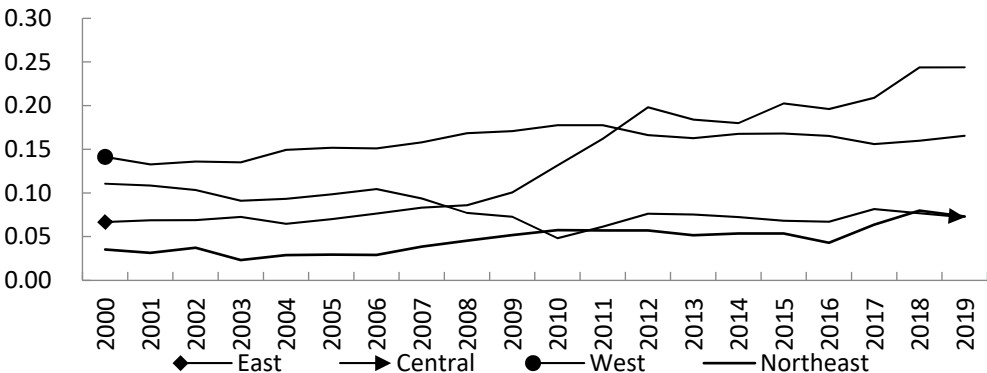

**Figure 4.** Inter-regional differences in the level of economic and social development quality in the four regions.

### 5.3. Inter-Regional Differences Expanding Rapidly, East–West Differences Still High

Based on Dagum's Gini coefficient decomposition method, Figure 5 illustrates the evolutionary trend of inter-regional differences in the level of high-quality economic and social development. During the sample period, the differences between the eastern and central regions, the eastern and western regions, and the eastern and northeastern regions showed a gradual increase, with the average annual increases in regional difference being 0.72%, 0.83%, and 1.01%, respectively. The difference between the central and western regions and between the western and northeastern regions fluctuated between 0.17 and 0.20. The difference between the central region and the northeastern region has been slowly declining, with an average annual rate of decline of 0.02%. Between 2000 and 2004, the inter-regional differences between the West, the East and the Northeast were at the same level, and at this time the Northeast was still at a level of economic and social development comparable to that of the East. Between 2005 and 2019, Dagum's Gini coefficient declined rapidly in the western and northeastern regions, while Dagum's Gini coefficient grew rapidly in the western and eastern regions at an annual rate of 0.83% and declined significantly in the central and northeastern regions. The largest inter-regional differences in 2019 were between the eastern and western regions, followed by the eastern and northeastern regions, the eastern and central regions, the central and western regions, the northeastern and western regions, and the central and northeastern regions.

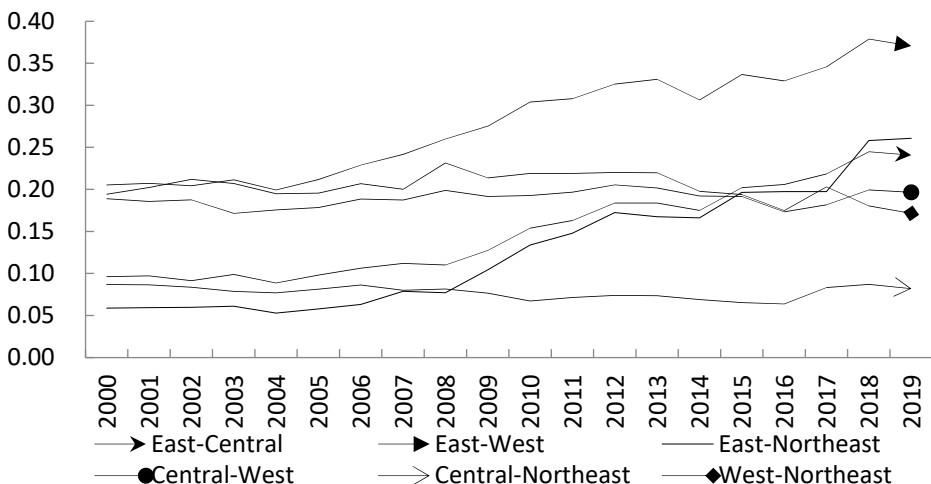

**Figure 5.** Inter-regional differences in the level of quality economic and social development in the four regions.

### 5.4. Inter-Regional Differences: The Main Driver of Differences in Development Quality

To reveal the drivers of the differences in the level of economic and social development quality in the four regions, this study measures the contribution of the three main drivers, as shown in Table 2. The average annual contribution of intra-regional, inter-regional, and super-variable densities during the sample period was 21.44%, 68.95%, and 9.61%, respectively. The drivers of variation in the level of economic and social development quality are, in order, inter-regional variation, intra-regional variation, and hypervariable density, with over 65% of the contribution of regional variation being caused by inter-regional variation, while intra-regional variation showed a fluctuating upward evolutionary trend and remained a constant source of regional variation, while the contribution of hypervariable density was stable with values between 6% and 15%.

**Table 2.** Drivers of differences in the level of quality economic and social development in the four regions.

| Year | Intra-Regional Variation | Contribution Rate/% | Inter-Regional Differences | Contribution Rate/% | Ultra-Variable Density | Contribution Rate/% |
|------|------|------|------|------|------|------|
| 2000 | 0.030 | 20.932 | 0.091 | 64.119 | 0.021 | 14.949 |
| 2001 | 0.029 | 20.246 | 0.094 | 66.752 | 0.018 | 13.002 |
| 2002 | 0.029 | 20.457 | 0.094 | 66.600 | 0.018 | 12.943 |
| 2003 | 0.029 | 20.314 | 0.096 | 68.509 | 0.016 | 11.177 |
| 2004 | 0.030 | 21.701 | 0.090 | 65.730 | 0.017 | 12.569 |
| 2005 | 0.031 | 21.454 | 0.096 | 66.977 | 0.016 | 11.569 |
| 2006 | 0.031 | 20.666 | 0.104 | 68.776 | 0.016 | 10.558 |
| 2007 | 0.033 | 20.778 | 0.109 | 69.632 | 0.015 | 9.590 |
| 2008 | 0.033 | 20.034 | 0.118 | 71.466 | 0.014 | 8.500 |
| 2009 | 0.035 | 20.369 | 0.125 | 72.047 | 0.013 | 7.583 |
| 2010 | 0.039 | 20.563 | 0.138 | 72.517 | 0.013 | 6.919 |
| 2011 | 0.044 | 22.000 | 0.138 | 69.362 | 0.017 | 8.637 |
| 2012 | 0.048 | 22.490 | 0.147 | 68.633 | 0.019 | 8.877 |
| 2013 | 0.046 | 21.574 | 0.151 | 70.929 | 0.016 | 7.496 |
| 2014 | 0.046 | 22.681 | 0.137 | 67.676 | 0.019 | 9.643 |
| 2015 | 0.049 | 22.476 | 0.153 | 69.536 | 0.018 | 7.988 |
| 2016 | 0.048 | 22.620 | 0.150 | 70.263 | 0.015 | 7.188 |
| 2017 | 0.050 | 22.137 | 0.160 | 70.891 | 0.016 | 6.972 |
| 2018 | 0.056 | 22.401 | 0.175 | 69.870 | 0.019 | 7.730 |
| 2019 | 0.056 | 22.849 | 0.169 | 68.695 | 0.021 | 8.456 |

## 6. Conclusions and Discussion

### 6.1. Conclusions

In terms of overall characteristics, the high-quality development of China's economy and society in the period 2000–2019 shows a strong complexity in terms of temporal and spatial dimensions, while the comprehensive index of development trend remained stable and positive. The eastern region has shown a strong horse-trading effect, the central region has emerged strongly, the western region has achieved significant results in its development, and the northeastern region has accelerated its pace of development. Analysis of the five-dimensional indices shows that the imbalance in the quality of inter-provincial economic and social development is primarily reflected in the three dimensions of economic dynamism, innovation capacity, and safety and security, contributing to the formation of higher composite indices in provinces such as Jiangsu, Guangdong, and Shandong. The two-dimensional indices of people's well-being and green ecology show a relatively good regional balance, with small inter-provincial differences. In terms of overall differences, the composite index shows an unbalanced trend of high in the east, flat in the middle, and low in the west, and the contradiction of unbalanced and insufficient development remains prominent. During the sample period, the disparities in China's economic and social development followed a fluctuating upward trend within a narrow range; the average annual increase is 0.525% and does not show a regular increasing trend, while the overall regional coordination is weak. Dagum's Gini coefficient remained within a reasonable threshold range. In terms of regional differences, the four major regions were at different levels of economic and social development quality during the sample period and showed significantly different evolutionary trends. Taking 2007 as a watershed year, Dagum's Gini coefficients for the western and central regions grew rapidly in the first period, while the northeastern and western regions maintained an average annual growth rate of 0.46% and 0.49,% respectively, in the second period. Dagum's Gini coefficients for the central region decreased with fluctuations, while the eastern region maintained a high average annual growth rate of 2% (i.e., intra-regional differences followed a widening trend). The contribution of inter-regional differences to regional differences was over 65%, while intra-regional differences contributed about 22%, indicating that the task of regional development and reform remains quite difficult.

### 6.2. Discussion and Outlook

Looking ahead to the 14th Five-Year Plan and beyond, the spatial layout of the level of economic and social development will not change, with the eastern region leading, the central region remaining flat, and the western and northeastern regions catching up. The resilience and tension of development formed by the long-term horse-power effect in the developed coastal regions will remain strong and maintain a relative advantage over the central and western regions. In view of this, it is necessary to strengthen the co-ordination of regional strategies in accordance with the changing situation and to take effective measures to reduce regional differences and promote the formation of a new pattern of regional development with relative balance and complementary advantages.

The eastern region needs to focus on enhancing its international competitiveness, global influence, and sustainable development, giving full play to its role as a model, leader, and driver of the double cycle, while taking the lead in modernization. The central region needs to accelerate its rise, build a modern industrial system supported by advanced manufacturing industries, play a more important role in the double cycle, and promote a high level of inland openness. The western region needs to speed up the formation of a new pattern of great protection, great openness, and high-quality development, as well as constantly enhance its ability to become integrated into the domestic cycle and give full play to the role of a double cycle node. The Northeast needs to combine economic relief with transformation, upgrading, and institutional restructuring to maintain national defense, food, ecological, energy, and industrial security, and to improve its level and ability to participate in the double cycle.

As major cities and urban agglomerations within the four regions have different development bases and are developing at different rates, intra-regional differences are likely to continue to widen, calling for regional policies to take a better role in promoting coordinated regional development, improving the spatial allocation of resource factors, achieving staggered regional development and organic integration, and enhancing the coordination of the four regions in the overall national development picture. First, we should follow the law of regional development and promote the flow of production factors and resource concentration in areas with good economic development conditions and high efficiency in the use of factors, so that they can better play the function of regional growth poles. Other regions should strengthen the functions of guaranteeing food, ecological, and border security, complementing and supplementing each other's strengths. Second, we should further promote the collaborative development of Beijing, Tianjin, and Hebei; the development of the Yangtze River Economic Belt; the construction of the Guangdong–Hong Kong–Macao Greater Bay Area; the integrated development of the Yangtze River Delta; and the ecological protection and high-quality development of the Yellow River Basin. Major regional strategies should also be promoted to make new breakthroughs. Finally, we should strengthen the advantages of regional central cities and city clusters in terms of gathering resources and economies of scale, build them into hubs for optimal allocation of factors, service platforms for high-quality industrial development and innovation radiation centers, and forge them into new sources of growth power for regional development.

**Author Contributions:** Supervision, project administration, funding acquisition, Q.M.; conceptualization, methodology, software, formal analysis, resources, data curation, writing—original draft preparation, M.B.; writing—review and editing, L.L. All authors have read and agreed to the published version of the manuscript.

**Funding:** This research was funded by the national natural science foundation of China (No. 71863033).

**Institutional Review Board Statement:** Not applicable.

**Informed Consent Statement:** Not applicable.

**Data Availability Statement:** The data came from the following website: (1) http://www.stats.gov.cn/tjsj/ndsj/ (accessed on 1 March 2021). (2) https://data.cnki.net/Yearbook/Navi?type=type&code=A (accessed on 1 March 2021).

**Conflicts of Interest:** The authors declare no conflict of interest.

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
