# Peer review of "Study on the Dynamic Evolution and Regional Differences of the Level of High-Quality Economic and Social Development in China"

_sustainability, doi:10.3390/su15010382_

Round 1
Reviewer 1 Report
The article is relevant, logically structured, the goals and objectives set in the article have been achieved.
The article is of scientific and practical interest. It contains the results of modeling based on a long dynamic series of indicators on the subject of research on the example of the Chinese economy. The conclusions and recommendations of the authors are based on the theoretical basis of the existing problems, as well as on the author's calculations, are consistent with the results of previous studies, supplement and develop them.
The article has a scientific novelty.
Recommendations for the revision of the article:
1. In the abstract and introduction, clearly highlight the purpose of the study.
2. Highlight the hypothesis of the study and indicate in the final part whether it was confirmed or refuted during the study.
3. In the Discussion section, show how the results of the study on the Chinese economy are consistent with previously conducted similar studies in other countries.
Author Response
We really appreciate your professional evaluation of our articles. As you are concerned about, there are several issues that need to be addressed. Based on your suggestions, we have made corrections to previous manuscripts, which are as follows:
- The purpose of the study has been clearly highlighted in the abstract and introduction.
- This paper does not design assumptions, but evaluates the high-quality economic development of various regions in China through high-quality development indicators.

Reviewer 2 Report
This study is compelling in its attempt to build a development evaluation index system oriented to high-quality economic and social development in China.
Figures and table are well performed, as well as the case study analysis.
The research is China-oriented and I appreciate so much all sections but I feel there is a lack of international perspective.
As the authors said, "High-quality development has become a fundamental requirement for China to determine its ideas related to development, formulate economic policies, and implement macro-control in the present and future. As a consequence I recommend to add the following themes in the section "Literature Review and Comments" as related references in order to improve the paper for the international readership of this journal.
1) In Western countries high-quality developments have provoked several distorsions references some examples of global cities:
- 2020. Alpha City: How London Was Captured by the Super-Rich. London: Verso
- 2019. Regenerating Bilbao: From ‘productive industries’ to ‘productive services’. Territorio, 89, 145-154. doi: 10.3280/TR2019-089019
- 2019. Capital City. Gentrification and the real estate state. London-New York: Verso
- 2018. Handbook of gentrification studies. Cheltenham-Northampton: Edward Elgar
- 2017. The icon project: architecture, cities and capitalist globalization. New York: Oxford University Press
2) The need to reflect on the spaces in which high-quality developments take place, i.e. brownfields/urban voids:
- 2022. Brownfield infrastructures. In: The Elgar Companion to Urban Infrastructure Governance, pp. 165-180
- 2020, A Glossary of Urban Voids, JOVIS Verlag, Berlin
3) Mention solutions undertaken by cities for providing economic vitality, innovation efficiency, green development, people’s life, and social harmony:
- 2020. Changing the urban design of cities for health: The superblock model. Environment International, 134: 105132
This is why I require a new version of this interesting paper.
Author Response
We really appreciate your professional evaluation of our articles. As you are concerned about, there are several issues that need to be addressed. Based on your suggestions, we have made corrections to previous manuscripts, Due to limited access rights, we have updated the literature in which you have commented, which can be viewed in the manuscript.

Round 2
Reviewer 2 Report
The paper can be published